# Anthropogenic shift towards higher risk of flash drought over China

Xing Yuan [1,2]*, Linying Wang[2], Peili Wu [3], Peng Ji[2,4], Justin Sheffield[5] & Miao Zhang[2,4]

Flash droughts refer to a type of droughts that have rapid intensification without sufficient early warning. To date, how will the flash drought risk change in a warming future climate remains unknown due to a diversity of flash drought definition, unclear role of anthropogenic fingerprints, and uncertain socioeconomic development. Here we propose a new method for explicitly characterizing flash drought events, and find that the exposure risk over China will increase by about 23% ± 11% during the middle of this century under a socioeconomic scenario with medium challenge. Optimal fingerprinting shows that anthropogenic climate change induced by the increased greenhouse gas concentrations accounts for 77% ± 26% of the upward trend of flash drought frequency, and population increase is also an important factor for enhancing the exposure risk of flash drought over southernmost humid regions. Our results suggest that the traditional drought-prone regions would expand given the human-induced intensification of flash drought risk.

[1] School of Hydrology and Water Resources, Nanjing University of Information Science and Technology, Nanjing 210044, Jiangsu, China. [2] Key Laboratory of Regional Climate-Environment for Temperate East Asia (RCE-TEA), Institute of Atmospheric Physics, Chinese Academy of Sciences, Beijing 100029, China. [3] Met Office Hadley Centre, Exeter EX1 3PB, UK. [4] College of Earth and Planetary Sciences, University of Chinese Academy of Sciences, Beijing 100049, China. [5] Geography and Environment, University of Southampton, Southampton SO17 1BJ, UK. *email: xyuan@nuist.edu.cn

The Fifth Assessment Report of the Intergovernmental Panel on Climate Change concluded that there was a low confidence in detecting and attributing human impact on drought changes since the middle of the 20th century over global land areas due to internal climate variability[1], data scarcity and drought index variety[2,3], resulting in large uncertainty in future drought projection[1,4–6]. However, the chance of concurrent droughts and heatwaves has increased substantially with the influence of global warming[7–10], suggesting a type of drought with rapid onset as accompanied by heat waves, which is termed as flash drought[11–17]. Flash droughts have occurred frequently in recent years including those over the central USA in 2012[12], southern China in 2013[14], southern Africa in 2015[17], and northern USA in 2017[18]. Flash droughts raise the risk of food and water security, environment sustainability or even human mortality due to low predictability[12], and their potential for triggering compound extreme events with heatwaves or wildfires[7,19]. Therefore, there is an urgent need to investigate flash drought risk and its underlying drivers in a changing climate.

Similar to conventional droughts, flash droughts occur with atmospheric conditions of positive geopotential height anomaly[15] and downward atmospheric motion at sub-seasonal time scales, but high evapotranspiration (ET) reduces soil moisture rapidly, and causes damage without sufficient early warning. Due to the intrinsic nature of extreme events[20,21] and the uncertainty in simulating the hydrological cycle[22,23], detecting and attributing the human fingerprint (e.g., greenhouse gas emissions) on the changes of flash droughts at regional scale are therefore quite challenging. Moreover, projections of future changes are often carried out for droughts at seasonal, annual, or longer time scales[5,6,24], while flash droughts occur at sub-seasonal time scale from a few weeks to months. The response of flash drought risk to greenhouse warming therefore remains unknown.

Quantifying flash drought risk requires an objective identification of a flash drought event. Similar to conventional drought events[25], flash drought events are also subject to the processes of onset and recovery. Flash droughts could either evolve into seasonal droughts (e.g., the 2012 central USA summer drought), or terminate independently without connection with droughts at longer time scales. However, existing definitions do not explicitly consider drought intensification processes, or have neglected the recovery stage as well as the severity of a flash drought event[13,15,20]. A popular definition for flash drought events in the hydro-climate community is based on the joint distribution of positive temperature anomaly (e.g., heatwave) and soil moisture deficit[13,20,26], although the eco-hydrological community has different opinions from the perspective of flash drought impact[16]. The former essentially defines an event with concurrent heat extreme and dry conditions, but not necessarily a drought event. For a drought event (no matter conventional drought or flash drought), the system should reach a water deficit for a period of time. If a dry anomaly lasts for a very short period (which is common in the previous definitions[13,20,26]), it may not have any significant impacts on the ecosystem or the society. Moreover, the rapid intensification is also a key feature to distinguish flash droughts from conventional droughts in terms of their physical characteristics and impacts, and the identification of the severity and different drought stages facilitates early warning and risk assessment during the evolution of flash droughts.

Here, we propose a new flash drought definition based on soil moisture that can capture both flash (rapid intensification of a drought condition, e.g., rapid decline in soil moisture) and drought (under a certain soil moisture threshold for a period of time) conditions. Based on the new definition, we attribute historical change of flash droughts regarding human fingerprint, and project future exposure risk of flash droughts over China by

carrying out land surface model (LSM) ensemble simulations driven by multiple climate model simulations from the fifth Coupled Model Intercomparison Project (CMIP5). We find a significant increase in flash drought risk over China during the middle and end of this century, especially over southern China, where both human and ecosystems have high exposure and poor adaptability. The increasing flash drought risk is mainly caused by greenhouse gas-induced anthropogenic climate change, where both long-term warming and increasing rainfall variability lead to a drier but more variable soil condition over the flash drought hotspots. Moreover, population increase is another important factor for the increase of exposure risk. Our results suggest that non-traditional drought regions should also receive attention for drought adaptation, given the increasing risk of flash droughts in a changing climate.

## Results

**Definition and characteristics of flash drought events**. To identify a flash drought event, we consider both the rapid decline rate of soil moisture and the dry persistency in this study: the pentad (5 days) mean root-zone (top 1 m) soil moisture decreases from above 40th percentile to 20th percentile, with an average decline rate of no less than 5% in percentile for each pentad (e.g., June 30–July 14 in Fig. 1); if the declined soil moisture rises up to 20th percentile again, the drought terminates (e.g., July 15–19 in Fig. 1); and the drought should last for at least 3 pentads (15 days). The first two criterions describe the onset and recovery stages of a flash drought event. Although the recovery threshold could increase from 20th to 40th percentile, here the 20th percentile is chosen to exclude many events that can last for more than 3–6 months if the 30th or 40th percentile threshold is used, where those events should be regarded as seasonal droughts instead of flash droughts. The third criterion is the minimal time that the soil moisture remains below 40th percentile, which excludes those events that decrease from above 40th percentile rapidly down to 20th percentile within 10 days, but then recover up to 40th percentile suddenly.

For the case illustrated in Fig. 1, the duration for this flash drought event is 20 days (4 pentads). As compared with previous

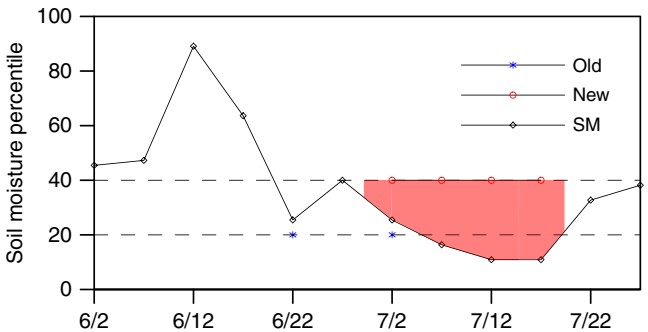

**Fig. 1** Illustration of the definition of a flash drought event. The black solid line shows 5 days mean soil moisture percentile (SM) for a grid point (112.75°E, 25.25°N) during the 2013 southern China flash drought. The dates in the horizontal axis are in the middle of the 5 days periods. The blue asterisks and red circles show flash droughts identified by concurrent heat and drought conditions (old; i.e., pentad mean surface air temperature anomaly larger than its standard deviation, and soil moisture percentile lower than 30%) and by the new definition used in this study (new; the pentad mean soil moisture decreases from above 40th percentile to 20th percentile, with an average decline rate of no less than 5% in percentile for each pentad; if the declined soil moisture rises up to 20th percentile again, the drought terminates; and the drought should last for at least three pentads), respectively

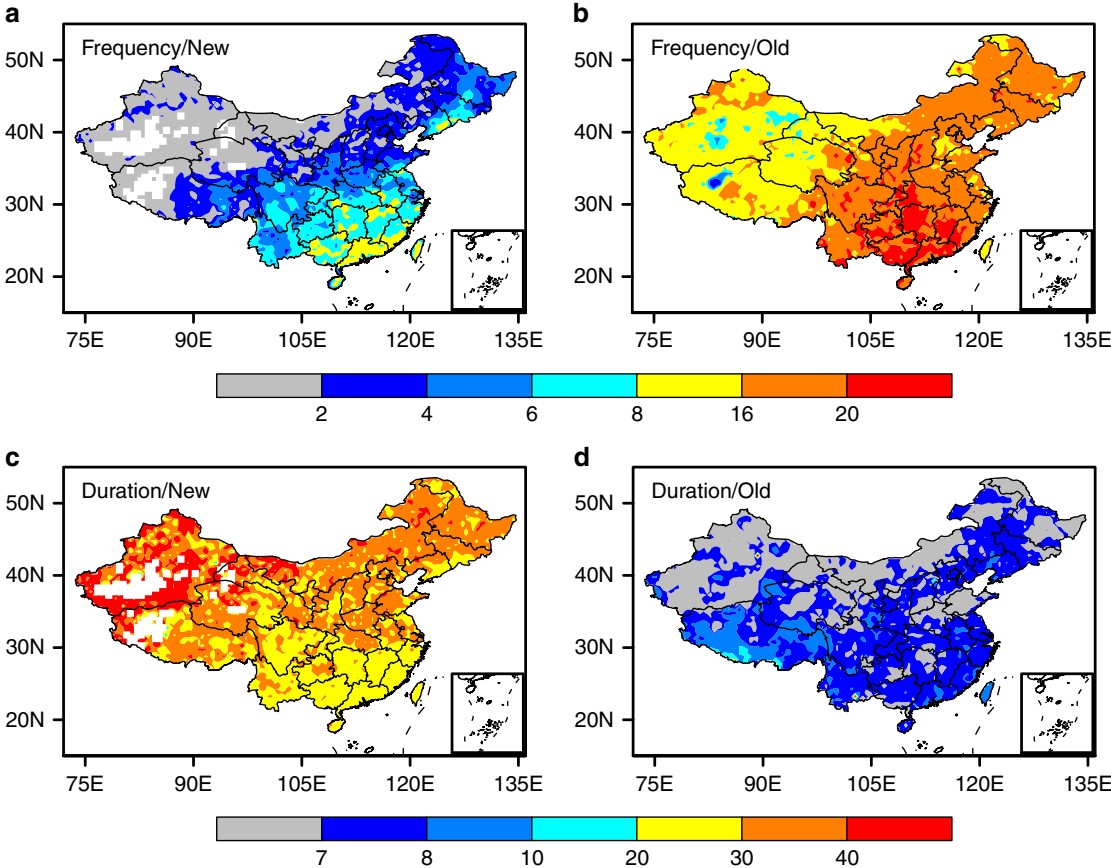

**Fig. 2** Comparison of mean frequency and duration of flash drought events based on two definitions. Mean results for the number of flash drought events per decade **a**, **b**, and durations (days) of flash drought events **c**, **d**. All statistics are based on the average results from three land surface model (i.e., CLM4.5, VIC, NoahMP) simulations driven by observed meteorological forcings during 1961–2005. Maps were created by using the NCAR Command Language (Version 6.3.0) [Software]. (2016). Boulder, Colorado: UCAR/NCAR/CISL/TDD. https://doi.org/10.5065/D6WD3XH5. And the maps were updated with a database provided by https://coding.net/u/huangynj/p/NCL-Chinamap/git/tree/master/database

concurrent heat and drought conditions[13,20] (blue asterisks in Fig. 1), the new definition reflects both the flash (decline rate of soil moisture percentile larger than 5%) and drought (soil moisture percentile <20%) conditions, while the previous definition may overestimate flash drought frequency with too short durations (usually only 5 days, e.g., June 20–24 in Fig. 1) that does not lead to any impact. Moreover, the severity of flash drought can also be explicitly estimated with the new definition, using the soil moisture percentile deficit during the flash drought event (red shaded area in Fig. 1).

With the new definition, the frequency, duration, and severity of flash drought events are calculated similar to those of conventional drought events[25], but at a higher temporal resolution. During a study period, the frequency is defined as the average number of flash drought events during the growing seasons per year, the mean duration is the average number of days during which an event lasts, and the mean severity is the mean accumulated soil moisture percentile deficits from the threshold of 40%. There is a higher chance for flash droughts over southern China (humid region) than over northern China (semiarid region), according to both the new and old definitions (Fig. 2a, b). There are more flash drought events based on the old definition (Fig. 2b), but their corresponding mean durations are very short, mostly around 7 days (Fig. 2d). In contrast, the mean durations of flash drought events based on the new definition are about 20–40 days (Fig. 2c), which is more reasonable in terms of drought impact[16].

**Future projection of flash drought risk.** To project future changes of flash drought risk, LSM ensemble simulations driven by historical and future meteorological conditions from CMIP5 climate models were performed, and the simulated soil moisture was used to calculate flash drought characteristics based on our new definition. The climate data from CMIP5 models were bias corrected (Supplementary Figs. 1–3; see the "Methods" section for details) before feeding into the LSMs, and the LSMs included the Community Land Model version 4.5 (CLM4.5)[27], the Variable Infiltration Capacity (VIC) model[28], and the Noah LSM with multiparameterization options (NoahMP)[29]. There are considerable uncertainties both from CMIP5 climate models and LSMs, where the uncertainty could be underestimated if using simulations with a single LSM (Supplementary Fig. 4). As compared with the LSM simulations driven by observed meteorological forcings, the CMIP5/ALL/LSM simulations underestimate the national-averaged flash drought frequency, but it is within the uncertainty of LSMs (Supplementary Fig. 5). Without using LSM ensemble simulations, the uncertainty range for the drought frequency might be underestimated by 21–44% (Supplementary Fig. 5).

Compared to the baseline period, the severity of flash drought is projected to increase significantly over southern China and part of northeastern China, and decrease slightly over part of northern China under the business as usual scenario (i.e., Representative Concentration Pathway 4.5 (RCP4.5)) during both the middle and the end of this century (Fig. 3a, b). On average, China flash

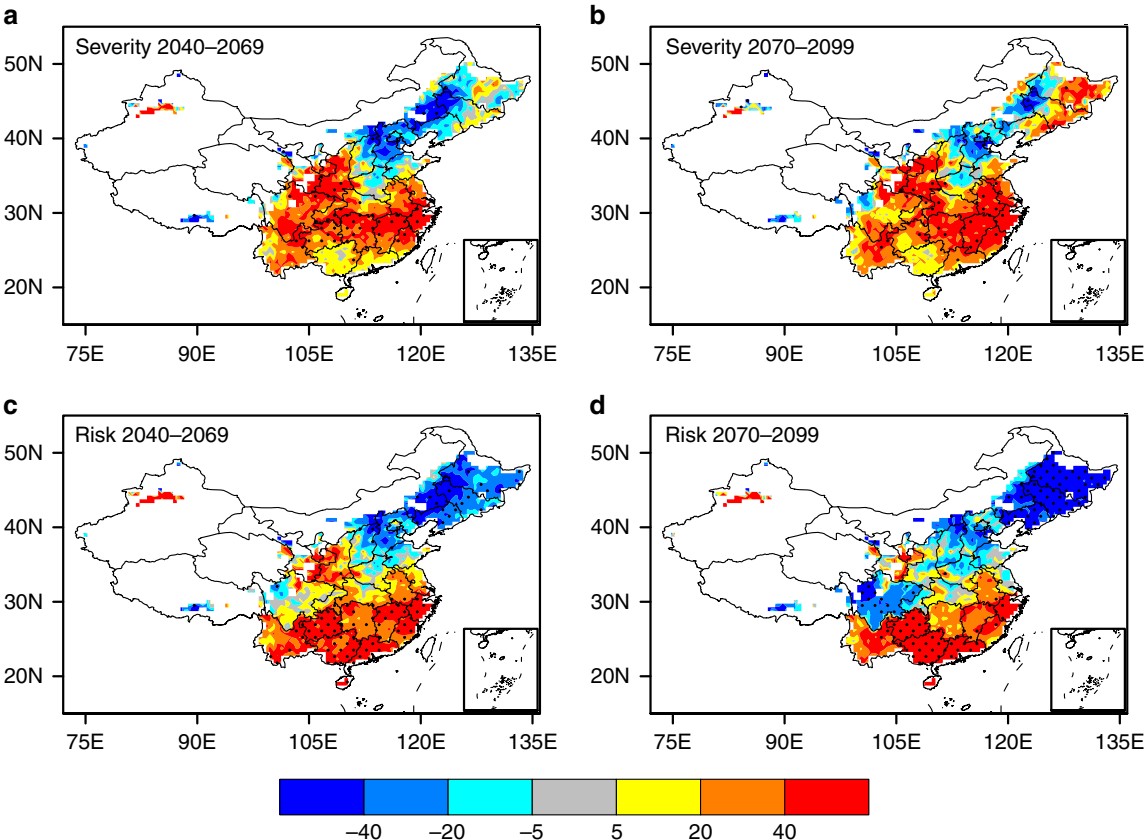

**Fig. 3** Future changes in flash drought severity and risk over China. Percentage changes in multimodel ensemble mean severity of flash drought from 1970–1999 to **a** 2040–2069 and **b** 2070–2099 simulated by multiple land surface models (i.e., CLM4.5, VIC, NoahMP) driven by multiple CMIP5 models under the RCP4.5 emission scenario (29 simulations in total, see Supplementary Table 1 for details). **c, d**, as for **a** and **b**, but for the changes in flash drought risk. Here, the risk is defined as exposure by multiplying population with drought duration (days*persons). The population scenarios for Shared Socioeconomic Pathways with medium challenges to mitigation and adaptation (SSP2) provided by National Climate Center at China Meteorological Administration was used for future projection. Regions with risk values <0.1 million days*persons per year were masked. Stippling denotes the regions where over 80% of the models agree on the sign of changes. All the statistics were calculated during the growing seasons (April–September). Maps were created by using the NCAR Command Language (Version 6.3.0) [Software]. (2016). Boulder, Colorado: UCAR/NCAR/CISL/TDD. https://doi.org/10.5065/D6WD3XH5. And the maps were updated with a database provided by https://coding.net/u/huangynj/p/NCL-Chinamap/git/tree/master/database

drought severity will increase by 18 ± 9% and 22 ± 12% during 2040–2069 and 2070–2099, respectively, with increases larger than 20% over most parts of southern China. The spatial patterns of the changes in frequency and duration are similar to the severity, where southern China will experience more frequent flash droughts with longer durations (Supplementary Fig. 6). In the flash drought projection, the uncertainty[30] from CMIP5 models is larger than that from LSMs, especially over northern China (Supplementary Fig. 7).

The flash drought exposure risk (days*persons) is estimated by multiplying annual population with annual duration of flash drought each year. Figure 3c, d show that future exposure to flash drought will increase by 23 ± 11% and 19 ± 16% averaged over China in the middle and end of this century. The hot spots for the increasing frequency and duration are located over Yangtze River basin (Supplementary Fig. 6), while the largest increases in flash drought risk (more than 40%) occur in the southernmost provinces (e.g., Guizhou, Guangxi, Guangdong; Fig. 3c, d), which suggests that climate change and population growth are the main reasons for the increasing flash drought risk over the north and south parts of southern China, respectively. For northern and northeastern China, future exposure risk of flash drought will decrease due to reductions in flash droughts and population, especially for the end of this century (Fig. 3c, d).

**Attribution of changes in flash drought risk**. To attribute flash drought changes, LSM simulations driven by CMIP5 climate simulations with different forcings were carried out (see the "Methods" section for details). There was an upward trend ($p$ < 0.05) for the frequency of flash drought events over China during 1961–2005, which was well captured by the CMIP5/ALL/LSM ensemble simulation ($p$ < 0.01) with both anthropogenic and natural forcings considered (Fig. 4a). However, the upward trend could not be reproduced by natural only forcings, where the ensemble simulation from CMIP5/NAT/LSM showed no trend. This suggests that anthropogenic climate change is mainly responsible for the increasing flash drought events over China. The upward trend from CMIP5/GHG/LSM ($p$ < 0.01) is similar to the all forcings simulation, indicating the dominant role of greenhouse gases in all anthropogenic forcings (Fig. 4a). The best estimates of scaling factors show that only the GHG signal is detectable, with a contribution of 77 ± 26% to the increase (Fig. 4b). These results are also consistent with the simulations of flash drought frequency, where GHGs play a dominant role in all anthropogenic forcings (Supplementary Fig. 8).

Besides the influence of GHGs emissions, the change of population is also an important factor for estimating the change of exposure risk of flash drought. Although the frequency and duration of flash drought increase in the future, the exposure risk

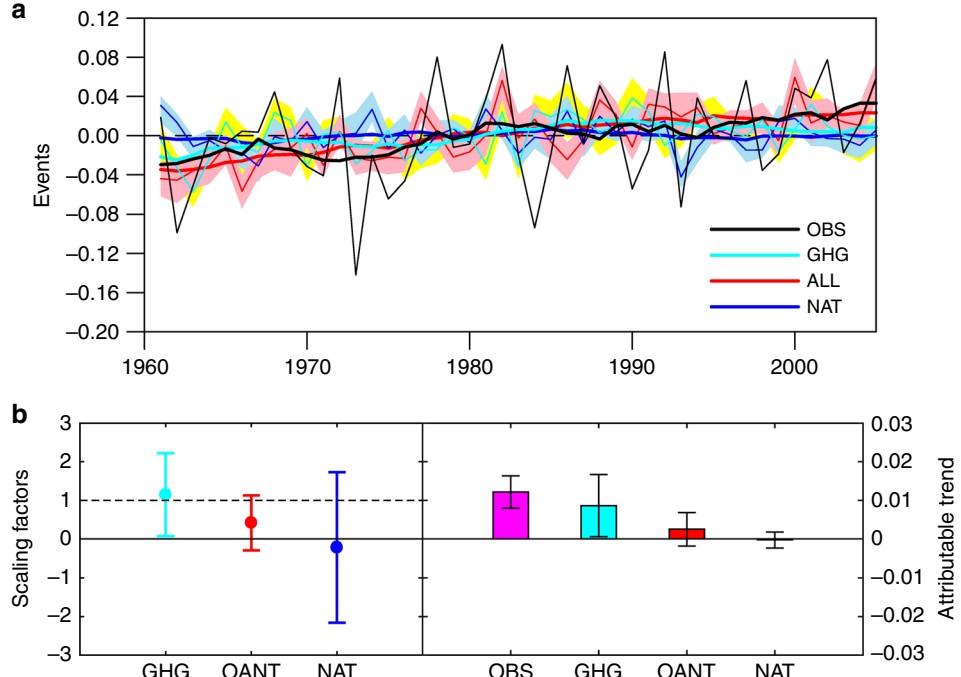

**Fig. 4** Attribution for historical change in flash drought frequency. **a** Observed and simulated anomalies of the number of flash drought events per year averaged over China. Black line shows the ensemble mean of OBS/CLM4.5, OBS/VIC, and OBS/NoahMP simulations, red, blue, and cyan lines show the land surface model ensemble simulations driven by CMIP5 climate model ensemble simulations with ALL, NAT, and GHG forcings, respectively (see Supplementary Table 1 for details). The thick lines are 10 years running means, and the pink, cyan, and yellow shadings display the 5–95% ranges of ALL, NAT, and GHG ensemble simulations respectively. **b** The best estimates of the scaling factors (left axis) and attributable increasing trends (right axis) from three-signal (GHG, OANT = ALL-GHG-NAT and NAT) analyses of flash drought changes for the period 1961–2005. Error bars indicate their corresponding 5–95% uncertainty ranges estimated via Monte Carlo simulations. All the statistics were calculated during the growing seasons (April–September)

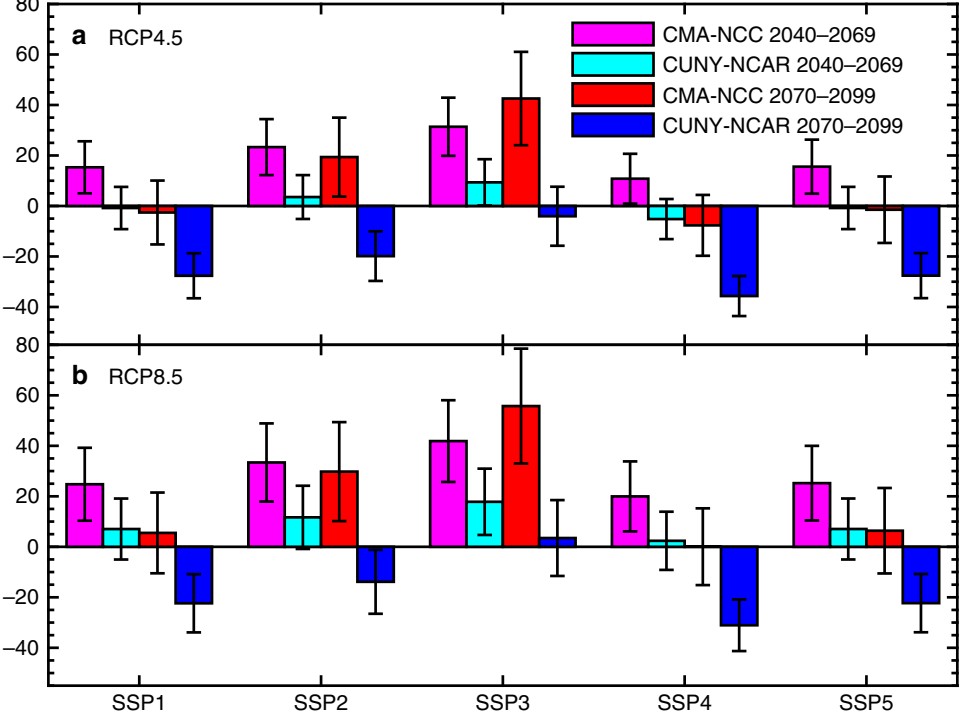

**Fig. 5** Future changes in flash drought risk averaged over China under different warming and socioeconomic scenarios. The risk changes under (**a**) RCP4.5 and (**b**) RCP8.5 (% relative to the 1970–1999 present-day level) scenarios based on 29 realizations with different combination of CMIP5 climate models and land surface models (see Supplementary Table 1 for details). Population under SSPs 1–5 from CMA-NCC and CUNY-NCAR were used to estimate exposure risk, where the former considered the recent two-child policy in China. All the statistics were calculated during the growing seasons (April–September). The black bars show 5–95% uncertainties which were estimated by using bootstrapping for 10000 times

over China may decrease significantly at the end of this century (blue bars in Fig. 5) based on a global population scenario data[31] (Supplementary Fig. 9). Since 2016, China has implemented the two-child policy to increase the birth rate, and a new population projection data[32] suggests that China's future population might be underestimated, especially at the end of this century (Supplementary Fig. 9). Based on the new population data, significant increase in the flash drought exposure risk over China is projected by the middle of this century (pink bars in Fig. 5a), although the increase becomes insignificant by the end of this century for a few SSPs (red bars in Fig. 5a). A higher warming scenario (RCP8.5) will further increase the risk by 9–11% according to the new CMA-NCC population data (Fig. 5b).

## Discussion

The results of this study have implications for understanding human influence on short-term hydrological extremes (e.g., flash drought), and how we quantify the impacts of future changes in both climate and socioeconomic development on the risk of extreme events. The impact of global warming on soil moisture drought[2,3,33] depends on whether the rising temperature causes ET to increase or decrease, which is uncertain because drought conditions usually inhibit ET especially in arid and semiarid regions. For flash drought, the rapid soil moisture decline should be a result of the intensification of ET driven by higher temperature, which is very common in humid and semi-humid regions where soil moisture can sustain high ET amount up to a few weeks. Our projection results suggest that GHG warming will increase growing seasonal mean ET and decrease soil moisture (Supplementary Fig. 10a, b, e–h), and therefore will increase flash droughts over southern China, although the growing seasonal mean precipitation will also increase (Supplementary Fig. 10c, d). Moreover, significant increase in short-term (i.e., 5-day mean) precipitation variability during growing seasons (Supplementary Fig. 11c, d) also contributes to the increase in the variability of ET and soil moisture (Supplementary Fig. 11e–h), raising a higher chance for short-term hydrological extremes including flash droughts over China.

The increasing trend in flash droughts over China is different from that over USA[13], but this is not due to the definition of flash droughts because our previous studies based on the definition of concurrent heat and dry anomalies also suggests upward trends in China[20]. While rigorous analysis based on comprehensive detection and attribution simulations should be carried out over USA to understand the difference, our preliminary speculation is that the difference may come from two sources: the focus periods and the internal climate variability are different between the studies over USA and China, which would influence the trend analysis; and the hotspots of flash droughts over USA (central Great Plains) experienced an increased soil moisture[13], while those over China (southern China) experienced significantly decreased soil moisture[20], again during different study periods. The different long-term trends in soil moisture may play an important role in altering flash drought trends, but the variability of hydroclimate variables (e.g., precipitation, temperature, and ET) at short time scales should also be investigated in detail.

Limiting the warming level as proposed by the Paris Agreement would have a nontrivial impact on reducing the risk of flash drought, but how to reduce the population exposure to flash droughts and how to develop adaptation and mitigation strategies are key to managing flash drought risk. As flash drought occurs more frequently especially in humid and semi-humid regions with a high population exposure, its risk will increase substantially in a warming climate with an accelerated urbanization and population growth, such as eastern China.

Flash droughts also have important implications for the ecosystem. Although a moderate drought may not necessarily hinder vegetation growth because there are less clouds and more solar radiation, flash droughts can cause serious damage to the ecosystems if they occur during critical stages of vegetation growth[13–15,17]. This situation would become even worse if a flash drought mixed with other extreme events[34,35]. Quantifying ecological impact of flash droughts is quite difficult because they are usually concurrent with heatwaves, and drought and heat extremes may have a synergistic adverse effect on the ecosystem. The flash droughts defined in this study do not explicitly consider heatwave conditions, but the heat and drought conditions may occur simultaneously for extreme cases, where a bivariate or multivariate (e.g., copula) analysis is needed for investigating their separate contributions and synergistic effect.

In addition, the rapid onset of flash droughts poses a great challenge for early warning. The reason that the 2012 central USA drought was so pernicious was because it started with a flash drought with a rapid onset and intensification which reduced time for preparation, and the flash drought was followed by a seasonal drought that lasted for 8 weeks. In this regard, the flash drought occurred at the onset stage of a seasonal drought[36]. Improving understanding of sub-seasonal predictability with consideration of vegetation–drought interactions and human interventions (e.g., agricultural practices), and developing climate–hydrology–human coupled prediction systems that utilize multiscale memory from the earth system (e.g., oceanic and land processes, or even human activities) would fundamentally increase our capability in managing risk of flash droughts, as well as those connected with seasonal droughts for both the society and environment.

## Methods

**Data**. Daily precipitation and surface air temperature observations[20] from 2474 China Meteorological Administration (CMA) stations (http://data.cma.cn/en) were interpolated into 0.5° resolution during 1959–2005. Other meteorological forcings including surface solar radiation, atmospheric humidity, wind speed, and pressure near surface were obtained by using a global observation dataset, the Climatic Research Unit-National Centers for Environmental Prediction version 7 (CRUNCEPv7) data[37] at 0.5° resolution during 1901–2005.

Daily precipitation and surface air temperature data from 11 CMIP5 model historical simulations[38] (Supplementary Table 1) that included all anthropogenic and natural forcings (ALL), greenhouse gases forcings (GHG), natural only forcings (NAT), or unforced pre-industrial situations (CTL), were used to drive three LSMs, i.e., CLM4.5[27], VIC[28], and NoahMP[29], to provide daily soil moisture data for the detection and attribution of historical changes in flash drought events. In fact, 13 CMIP5/ALL/LSM simulations were carried out, and 11 CMIP5 models were selected due to their capability in reproducing the upward trend for flash drought frequency over China for CLM4.5 and VIC, while seven CMIP5 models were selected for NoahMP (see Supplementary Table 1 for details). Daily precipitation and surface air temperature future projection data from the above 11 CMIP5 models were used to drive LSMs to provide daily soil moisture for projecting future flash drought changes. Both the simulations under RCP4.5 and RCP8.5 scenarios were used.

A global 1/8° gridded population projection data under different Shared Socioeconomic Pathways (SSP) scenarios during 2010–2100 provided by City University of New York and National Center for Atmospheric Research (CUNY-NCAR)[31] was aggregated into 0.5° over China. The original CUNY-NCAR population data was available every 10 years, and they were linearly interpolated into annual data. To better capture future population changes in China, a new annual dataset based on the 6th national census in 2010, as well as the newly released two-child policy in China (CMA-NCC)[32] was also used to estimate the population during 2010–2100. For the estimation of risk during 1970–1999, the population data in 2010 was used.

**Experimental design**. To estimate daily soil moisture as well as variation of flash droughts, the LSM simulations under different climate scenarios were performed. First, CRUNCEPv7 data during 1901–1958 was used to drive LSMs to provide an initial land surface condition for subsequent runs. Second, CMA precipitation and temperature observations during 1959–2005 were used to replace those in CRUNCEPv7 data, and the merged data was used for LSM simulations for two cycles from 1959 to 2005, with the land surface conditions at the end of the first cycle being used for the initial conditions for the simulations in the second cycle.

The results during 1961–2005 (dropping the 1959–1960 results for spin-up) in the second cycle were regarded as OBS/LSM simulations for the analysis in this paper. Third, CMIP5/LSM two-cycle runs similar to the OBS/LSM were carried out, by replacing CRUNCEPv7 precipitation and temperature during 1959–2005 with CMIP5 simulation data, and the corresponding 1961–2005 results in the second cycle were regarded as CMIP5/ALL/LSM, CMIP5/GHG/LSM, CMIP5/NAT/LSM simulations. The CMIP5/CTL/LSM experiments were conducted similarly, by repeatedly using CRUNCEPv7 data (except for precipitation and temperature) for the four sets of 45 years chunks for each CMIP5/CTL model simulations. These CMIP5/LSM historical simulations provided baseline information for the estimation of flash drought change, and were used for the detection and attribution analysis. Finally, RCP4.5 and RCP8.5 results from CMIP5 simulations were used to drive LSMs for future projections during 2006–2099, with initial conditions produced by CMIP5/ALL/LSM simulations at the end of 2005.

To reduce the bias from CMIP5 historical simulations, a quantile-mapping method[39] was used by adjusting the cumulative distribution functions (CDFs) from CMIP5/ALL simulations of precipitation and temperature to the observed CDFs during 1959–2005. GHG, NAT, and CTL simulations were also corrected based on the same CDFs from ALL simulations during 1959–2005. For the future projection experiments, the CMIP5/RCP4.5 and CMIP5/RCP8.5 outputs were corrected based on the equidistant CDF matching method[40] by accounting for distribution changes in historical (1959–2005) and projection (2006–2099) conditions. These bias corrections were conducted at monthly time scale, and the corrected monthly precipitation and temperature was applied to adjust daily CMIP5 outputs before being used to drive the LSMs for flash drought simulations. Supplementary Figs. 1–3 show that the bias correction method performed well during historical period and preserved the trends for future projections. Although all bias corrections use 1959–2005 as the baseline period for constructing historical distributions and correcting future projections, the soil moisture percentile estimations use 1961–2005 as the baseline period for both historical (1961–2005) attribution and future (2006–2099) projection of flash droughts, because the soil moisture data during 1959–1960 are dropped as LSM spin-up results.

**Detection and attribution method**. We applied the optimal fingerprint method[41] to analyze the anthropogenic climate change influence. The method regresses the observed changes ($y$) against the forced multimodel signal patterns ($X$) as: $Y = \beta X + \varepsilon$. In this study, three-signal analysis was conducted, where $X$ was expressed as GHG, OANT (ALL-GHG-NAT), and NAT forcings on non-overlapping 3-year averages of flash drought frequency. The scaling factors ($\beta$) represent the model simulated response patterns to match observations, which were estimated by the total least-squares method. $\varepsilon$ is the residual or the internal variability, and it was estimated using the CTL ensembles with 36 trunks from nine models (Supplementary Table 1). The uncertainties in the scaling factors give a measure of whether a particular external forcing is detected. If the 90% confidence interval of $\beta$ is above zero, the corresponding signal factor is claimed to be detected at a significance level of 5%. If the uncertainty range of $\beta$ also includes one, the attribution supports that the observed changes are consistent with the external forcings.

## Data availability

The CRUNCEP forcing data are available at UCAR website (https://svn-ccsm-inputdata.cgd.ucar.edu/trunk/inputdata/atm/datm7/). Daily precipitation and temperature observations from China Meteorological Administration (CMA) stations are available through http://data.cma.cn/en. The CMIP5 data was provided by the World Climate Research Program's Working Group on Coupled Modeling (http://cmip-pcmdi.llnl.gov/cmip5/availability.html). The CLM4.5 is available at CESM website (http://www.cesm.ucar.edu/models/cesm1.2/), and VIC and NoahMP are archived at https://vic.readthedocs.io/en/master/ and http://www.jsg.utexas.edu/noah-mp/.

## Code availability

Statistical methods are illustrated through text and figure captions. The analyzing data and drawing plots computer codes are in Fortran or NCAR Command Language (NCL) scripts. All the scripts are available upon request.

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

## Acknowledgements
We thank Prof. Tong Jiang at CMA/NCC for providing the gridded data for the projection of China population. We thank Prof. Luis Samaniego and an anonymous reviewer for their constructive comments. This work was supported by National Natural Science Foundation of China (41875105), National Key R&D Program of China (2018YFA0606002), and the Startup Foundation for Introducing Talent of NUIST. Peili Wu was supported by the UK-China Research & Innovation Partnership Fund through the Met Office Climate Science for Service Partnership (CSSP) China as part of the Newton Fund.

## Author contributions
X.Y. conceived and designed the study. X.Y., L.W. and P.J. conducted the simulations and performed the analyses, P.W. and J.S. provided critical insights on the results interpretation. X.Y. wrote the initial draft of the paper, with substantial contributions from all authors.

## Competing interests
The authors declare no competing interests.
