## [Peer Review File · Nature Communications]

Reviewers' comments:

Reviewer #1 (Remarks to the Author):

In general the manuscript is interesting and novel as it applied optimal fingerprinting to analyze rapid soil moisture depletion. However, the manuscript is also fundamentally flawed because of its lack of consideration for duration of events it coins as "flash droughts". The flash drought methodology improves considerably on that of Mo and Lettenmaier (2015, 2016); however, duration of the overall event is still not considered. Indeed, the authors frequently refer to the "short duration" of flash drought, while citing studies from Ford and Labosier (2017) and Otkin et al. (2018) that specifically argue flash drought is defined by its intensification and duration. That is, an event is a flash drought if it exhibits 1) rapid intensification and/or onset (i.e., the flash) and 2) sufficient duration to cause some physical or socioeconomic problem (i.e., the drought). The authors use events such as the 2012 central U.S. drought as evidence for the socioeconomic impact of flash drought; however, they fail to mention that the 2012 event lasted for more than 8 weeks. The reason the 2012 event was so pernicious was because it exhibited both 1) rapid onset/intensification, reducing time for preparation AND 2) sufficient intensity and duration to significantly diminish crop productivity and yield. Therefore, it is paramount in flash drought studies that one use both duration and intensification rate to identify these events. I strongly encourage the authors incorporate these changes to their methodology, which will enhance the impact of their findings.

Reviewer #2 (Remarks to the Author):

This paper investigates trends in the (frequency of) flash droughts over China, which the authors predict will increase by 30% (40% in the South) over the second half of this century relative to the base period 1959-2005 (although this is not entirely clear as they mention data for the period 1901-2005 as well – however given that there are essentially no precipitation observations over China suitable for gridding prior to about 1960, it seems more likely that their base period was 1901-2005). They come to this conclusion using future climate simulations from 11 CMIP5 climate simulations.

This is not a very well written or conceived paper, and frankly, I don't think it warrants a very detailed review. In my view, the main issues are:

a) Poor description of the methods. For instance, what do they do about bias correcting the GCM outputs? Anyone who has worked with GCM output knows that the biases are huge (typically at least 1-2 degrees C for instance on average, and several 10s of percent for precipitation). So if that bias isn't corrected, it will dominate the results. Just how that is done is critical, and needs to be in the methods section, not buried in a supplement.

b) They apparently create their own, new definition of flash drought specifically for this paper. Why? Two recent papers (Mo and Lettenmaier, 2015, which they reference, and Mo and Lettenmaier, 2016, which they don't) develop definitions of two types of flash droughts (heat wave and precipitation deficit) and discuss the physics that go along with them. What is the motivation for developing another definition, the only effect of which, as nearly as I can tell, is to make it impossible to compare with the U.S. results in these two papers? Furthermore, Mo and Lettenmaier (2015) find that heat wave flash droughts have been trending down over the last 90 years or so in the U.S., and precipitation deficit events are unchanged. What are the particulars of the Chinese data set(s) that lead to increases?

c) Apparently they haven't investigated trends in the historic data? That seems to me to be the place one would start, rather than investigating climate model output. With climate model output, you know things are going to change – that's pretty much a given – but whether the results make any sense is indeterminate if you don't at least do an analysis of the historic record.

d) The last author (Sheffield) published a paper a few years ago arguing no change in droughts globally over the historic record. Yes, these are flash droughts, and he and colleagues investigated more traditional large area persistent droughts. But still, I would expect some interpretation of why the flash drought trends over China are in such strong contrast. Furthermore, while they do reference the 2012 Sheffield et al paper, it is in the context of an introductory statement about "large uncertainty in future drought projections" – this even though the 2012 paper deals with historic droughts "over the last 60 years" and to my recollection, has nothing to do with future projections.

Taken together, my recommendation is that this paper should not be seriously considered by Nature Communications. Frankly, I don't think it should have been sent out for review in the first place. The authors would be far better advised to submit their work to a quality disciplinary journal that has a history of publishing in the area, e.g., JHM or J Clim.

Reviewer #3 (Remarks to the Author):

1. General Comments This manuscript presents a new method to characterise flash drought events (i.e., rapid onset, short duration, co-evolving with a heatwave, severe impacts) and to estimate the exposure risk. This method is applied in China because the propensity of semi-humid and humid regions to endure this kind of hazard. The authors claim that this type of event will increase 30% by the second half of this century under a medium socioeconomic challenge scenario increase. Using the fingerprinting approach, the authors attribute that 80% of the expected changes are induced by the increased greenhouse gas concentrations. The subject covered by this study is a highly relevant research topic having broad scientific and public interest, hence, it is highly suitable for this journal. In the present manuscript, however, there are many key points that have to be clarified before publication.

2. Specific Comments

- My major concern with this study is the lack of uncertainty estimates for all major results. It is highly recognised in the literature (see the 2018 NCC 1 article authored by the last co-author, ref. nr. 6) that the SMI is highly uncertain due to the large influence of the land surface parameterizations. For this reason is highly recommended to use several LSM to estimate the uncertainty bounds of the SMI estimates (as done in the previous reference). In the present case, the authors only use CLM4.5 (see L.70) to estimate the SMI using CMIP5 forcings. In this way, the authors can estimate the GCM variability but not the LSM variability and hence the results may be over or under-estimated, depending of the bias of the current LSM. Therefore, I strongly suggest the authors to use at least two more LSM for generating a super ensemble. Then, estimate the standard errors of the major results.

- The link with the heatwaves (part of the definition of flash droughts) is not clear based on what us written. Please elaborate (L147, L240ff). How temperature is taken into account for the definition of a flash drought event? Or is not taken? If not, then the introduction is misleading (L35 ff).

- I strongly recommend to use a trend-preserving bias correction (see S. Hempel et al. doi:10.5194/esd-4-219-2013.) instead of the equidistant CDF matching method (L312). The method used my have changed the trends of P and T in the CMIP5 models and hence may alter the results of this study.

- Sampling uncertainty (variability in the model-simulated response) is key for the optimal fingerprinting approach (see Allen and Scott, 2003 10.1007/ s00382-003-0313-9). In the present case, I consider that the SMI uncertainty is totally underestimated by the use of a single LSM (CLM4.5). Please investigate, how the inclusion of more LSMs would affect the 80% estimate.

- It would be interesting to know what is the uncertainty contribution of GCMs and LSMs on the flash drought occurrences. This, of course, only can be done by including more LSM in this study.

3. Final Remarks Based on the comments mentioned above and bearing in mind the NCOMMS publishing standards for a research article, I recommend to return this manuscript to the Authors for major revisions.

Responses to the comments from Reviewer #1

We thank the reviewer for the critical review. The thoughtful comments have helped improve the manuscript. The reviewer's comments are italicized and our responses immediately follow.

In general the manuscript is interesting and novel as it applied optimal fingerprinting to analyze rapid soil moisture depletion. However, the manuscript is also fundamentally flawed because of its lack of consideration for duration of events it coins as "flash droughts". The flash drought methodology improves considerably on that of Mo and Lettenmaier (2015, 2016); however, duration of the overall event is still not considered. Indeed, the authors frequently refer to the "short duration" of flash drought, while citing studies from Ford and Labosier (2017) and Otkin et al. (2018) that specifically argue flash drought is defined by its intensification and duration. That is, an event is a flash drought if it exhibits 1) rapid intensification and/or onset (i.e., the flash) and 2) sufficient duration to cause some physical or socioeconomic problem (i.e., the drought). The authors use events such as the 2012 central U.S. drought as evidence for the socioeconomic impact of flash drought; however, they fail to mention that the 2012 event lasted for more than 8 weeks. The reason the 2012 event was so pernicious was because it exhibited both 1) rapid onset/intensification, reducing time for preparation AND 2) sufficient intensity and duration to significantly diminish crop productivity and yield. Therefore, it is paramount in flash drought studies that one use both duration and intensification rate to identify these events. I strongly encourage the authors incorporate these changes to their methodology, which will enhance the impact of their findings.

Response: We would like to thank the reviewer for the positive and constructive comments. We totally agree with the reviewer that *"an event is a flash drought if it exhibits 1) rapid intensification and/or onset (i.e., the flash) and 2) sufficient duration to cause some physical or socioeconomic problem (i.e., the drought)"*, and it is also the motivation for us to revise the definition of Mo and Lettenmaier (2015, 2016). In fact, we started our flash drought study by following Mo and Lettenmaier (2015), and identified an increasing trend for China flash drought during past three decades (Wang and Yuan et al., 2016). Then we investigated the causes of the increasing trends and explored the future changes for the flash droughts in China, but found the previous definition (i.e., concurrent heat and dry anomaly) results in many events with too short

durations (e.g., 5 days) to have any impact, and it did not explicitly characterize the rapid evolution of flash droughts. Therefore, in this study, we have proposed a definition that can consider both the duration and intensification rate.

1) To clarify the flash drought definition, we have now moved those statements from the Methods section to the beginning of the Results section, and explained the motivation of proposing a new definition in the revised manuscript as follows:

“Definition and characteristics of flash drought events. A popular definition for flash drought events in the hydroclimate community is based on the joint distribution of positive temperature anomaly (e.g., heatwave) and soil moisture deficit^{13,20,26}, although the ecohydrological community has different opinions from the perspective of flash drought impact¹⁶. The former essentially defines an event with concurrent heat extreme and dry conditions, but not necessarily a drought event. For a drought event (no matter conventional drought or flash drought), the system should reach a water deficit for a period of time. If a dry anomaly lasts for a very short period (which is common in the previous definitions^{13,20,26}), it may not have any significant impacts on the ecosystem or the society. Therefore, we consider both the rapid decline rate of soil moisture and the dry persistency in this study: 1) when the percentile for pentad (five-days) mean soil moisture is lower than 40%, a flash drought onset is identified if the soil moisture percentile continuously decreases into 20%, with an average decline rate of no less than 5% for each pentad (e.g., June 30-July 9 in Fig. 1); and 2) if the average decline rate is less than 5% per pentad, the flash drought terminates (e.g., July 10-July 14 in Fig. 1). These two stages are regarded as onset and recovery stages of a flash drought event. For the case illustrated in Fig. 1, the duration for this flash drought event is 15 days (3 pentads). As compared with previous concurrent heat and drought conditions^{13,20} (blue asterisks in Fig. 1), the new definition reflects both the “flash” (decline rate of soil moisture percentile larger than 5%) and “drought” (soil moisture percentile less than 20%) conditions, while the previous definition may overestimate flash drought frequency with too short durations (usually only 5 days, e.g., June 20-24 in Fig. 1) that does not lead to any impact. Moreover, the severity of flash drought can also be explicitly estimated with the new definition, using the soil moisture percentile deficit during the flash drought event (red shaded area in Fig. 1).” (L75-L98 in the revised manuscript)

Figure 1. Illustration of the definition of a flash drought event. The black solid line shows five-days mean soil moisture percentile (SM) for a grid point (112.75°E, 25.25°N) during the 2013 southern China flash drought. The dates in the horizontal axis are in the middle of the five-days periods. The blue asterisks and red circles show flash droughts identified by concurrent heat and drought conditions (old; i.e., pentad mean surface air temperature anomaly larger than its standard deviation, and soil moisture percentile lower than 30%) and by the new definition used in this study (new; see text for details), respectively.

2) All analysis in this study DOES consider the duration of flash drought. We have now clarified them in the revised manuscript as follows:

“Frequency, duration and severity of flash drought events are calculated similar to those of conventional drought events²⁵, but at a higher temporal resolution. During a study period, the frequency is defined as the average number of flash drought events during the growing seasons per year, the mean duration is the average number of days during which an event lasts, and the mean severity is the mean accumulated soil moisture percentile deficits from the threshold of 40%. There is a higher chance for flash droughts over southern China (humid region) than over northern China (semiarid region), according to both the new and old definitions (Figs. 2a-2b). There are more flash drought events based on the old definition (Fig. 2b), but their corresponding mean durations are very short, mostly around 7 days (Fig. 2d). In contrast, the mean durations of

flash drought events based on the new definition are about 15-20 days (Fig. 2c), which is more reasonable in terms of drought impact¹⁶.” (L99-L109)

Figure 2. Comparison of mean frequency and duration of flash drought events based on two definitions. Mean results for the number of flash drought events per decade (a)-(b), and durations (days) of flash drought events (c)-(d). All statistics are based on the average results from three land surface model (i.e., CLM4.5, VIC, NoahMP) simulations driven by observed meteorological forcings during 1961-2005.

“The spatial patterns of the changes in frequency and duration are similar to the severity, where southern China will experience more frequent flash droughts with longer durations (Supplementary Fig. 6).” (L131-L133)

Figure S6. The same as Figure 3, but for the relative changes (%) in frequency and duration of flash drought event. All the statistics were calculated during the growing seasons (April-September).

3) We would like to thank the reviewer for pointing out the causes of the severe impact of the 2012 central USA drought. We have now incorporated them in the Discussion section as follows:

“In addition, the rapid onset of flash droughts poses a great challenge for early warning. The reason that the 2012 central USA drought was so pernicious was because it started with a flash drought with a rapid onset and intensification which reduced time for preparation, and the flash drought was followed by a subseasonal drought that lasted for 8 weeks. In this regard, the flash drought occurred at the onset stage of a subseasonal drought³⁵. Improving understanding of sub-seasonal predictability with consideration of vegetation-drought interactions and human interventions (e.g., agricultural practices), and developing climate-hydrology-human coupled

prediction systems that utilize multiscale memory from the earth system (e.g., oceanic and land processes, or even human activities) would fundamentally increase our capability in managing risk of flash droughts as well as those connected with subseasonal or seasonal droughts for both the society and environment.” (L220-L230)

Responses to the comments from Reviewer #2

We thank the reviewer for the critical review. The reviewer's comments are italicized and our responses immediately follow.

This paper investigates trends in the (frequency of) flash droughts over China, which the authors predict will increase by 30% (40% in the South) over the second half of this century relative to the base period 1959-2005 (although this is not entirely clear as they mention data for the period 1901-2005 as well – however given that there are essentially no precipitation observations over China suitable for gridding prior to about 1960, it seems more likely that their base period was 1901-2005). They come to this conclusion using future climate simulations from 11 CMIP5 climate simulations.

Response: As illustrated in the caption of Figure 3 (Figure 1 in the previous version of the manuscript), the base period is 1970-1999, which is the same as many climate change studies that estimate 21st century changes as compared with those in the end of 20th century.

“Percentage changes in multimodel ensemble mean severity of flash drought from 1970-1999 to (a) 2040-2069 and (b) 2070-2099 simulated by...” (L446-L448 in the revised manuscript)

This is not a very well written or conceived paper, and frankly, I don't think it warrants a very detailed review. In my view, the main issues are:

a) Poor description of the methods. For instance, what do they do about bias correcting the GCM outputs? Anyone who has worked with GCM output knows that the biases are huge (typically at least 1-2 degrees C for instance on average, and several 10s of percent for precipitation). So if that bias isn't corrected, it will dominate the results. Just how that is done is critical, and needs to be in the methods section, not buried in a supplement.

Response: We agree with the reviewer that the bias correction is critical for climate change impact studies. For this study, we DID clarify the bias correction in the Methods section (the last paragraph in Experimental design in the Methods section), and all our LSM simulations driven by CMIP5 models are based on the bias-corrected climate data. To illustrate the importance of

bias correction, we have now plotted the CMIP5 results before and after bias correction in Supplementary Figs. 1-3 as follows:

“To reduce the bias from CMIP5 historical simulations, a quantile-mapping method³⁸ was used by adjusting the cumulative distribution functions (CDFs) from CMIP5/ALL simulations of precipitation and temperature to the observed CDFs. NAT and CTL simulations were also corrected based on the same CDFs from ALL simulations. For the future projection experiments, the CMIP5/RCP4.5 and CMIP5/RCP8.5 outputs were corrected based on the equidistant CDF matching method³⁹ by accounting for distribution changes in historical and projection conditions. These bias corrections were conducted at monthly time scale, and the corrected monthly precipitation and temperature was applied to adjust daily CMIP5 outputs before being used to drive the land surface model CLM4.5 for flash drought simulations. Supplementary Figs. 1-3 show that the bias correction method performed well during historical period and preserved the trends for future projections.” (L390-L400)

Figure S1. Comparison of original and bias-corrected CMIP5 July mean precipitation (mm/day) averaged over China during 1959-2005. Black lines are observations, solid and dashed red lines are original and bias-corrected CMIP5 precipitation. The equidistant CDF matching method (see Methods for details) was used for bias correction of monthly model data.

Figure S2. The same as Figure S1, but for surface air temperature (K).

Figure S3. An example of CMIP5 model data with or without bias correction for historical and future July precipitation (a; mm/day) and temperature (b; mm/day) averaged over China under ALL forcings (1959-2005) and RCP4.5 and RCP8.5 scenarios (2006-2099).

b) They apparently create their own, new definition of flash drought specifically for this paper. Why? Two recent papers (Mo and Lettenmaier, 2015, which they reference, and Mo and Lettenmaier, 2016, which they don't) develop definitions of two types of flash droughts (heat wave and precipitation deficit) and discuss the physics that go along with them. What is the motivation for developing another definition, the only effect of which, as nearly as I can tell, is to make it impossible to compare with the U.S. results in these two papers? Furthermore, Mo and Lettenmaier (2015) find that heat wave flash droughts have been trending down over the last 90

years or so in the U.S., and precipitation deficit events are unchanged. What are the particulars of the Chinese data set(s) that lead to increases?

Response:

- 1) Our previous studies suggest that both the heat wave flash droughts (Wang and Yuan et al., 2016) and precipitation deficit flash drought (Wang and Yuan, 2018) increased significantly over China, by following the definition of Mo and Lettenmaier (2015, 2016). However, this is not the motivation of proposing a new flash drought definition. In this study, we would like to investigate the causes of the increasing trends and explore the future changes for the flash droughts in China. We found the previous definition (i.e., concurrent heat and dry anomaly) results in many events with too short durations (e.g., 5 days) to have any impact, and it did not explicitly characterize the rapid evolution of a flash drought. Therefore, in this study, we have proposed a definition that can consider both the duration and intensification rate. For details, please see our response to the Reviewer #1 above for the motivation of developing a new definition, which has also been included in the revised version of the manuscript.

- 2) Although the question raised by the reviewer regarding the reason for the different changes in flash droughts over USA and China is not relevant to the main scope of this study, it is an interesting question. So, we have now included a brief discussion in the revised manuscript as follows:

“The increasing trend in flash droughts over China is different from that over USA¹³, but this is not due to the definition of flash droughts because our previous studies based on the definition of concurrent heat and dry anomaly also suggests upward trends in China²⁰. While rigorous analysis based on comprehensive detection and attribution simulations should be carried out over USA to understand the difference, our preliminary speculation is that the difference may come from two sources: 1) the focus periods and the internal climate variability are different between the studies over USA and China, which would influence the trend analysis; and 2) the hotspots of flash droughts over USA (central Great Plains) experienced an increased soil moisture¹³ while those over China (southern China) experienced significantly decreased soil moisture²⁰, again during

different study periods. The different long-term trends in soil moisture may play an important role in altering flash drought trends, but the variability of hydroclimate variables (e.g., precipitation, temperature and ET) at short time scales should also be investigated in details.” (L189-L201)

c) Apparently they haven't investigated trends in the historic data? That seems to me to be the place one would start, rather than investigating climate model output. With climate model output, you know things are going to change – that's pretty much a given – but whether the results make any sense is indeterminate if you don't at least do an analysis of the historic record.

Response: No, actually we DID investigate the trends in the historical data. Fig. 4a (Fig. 2a in the previous version) shows the changes of flash droughts during 1961-2005, and we have also compared the CMIP5-driven results (colored lines and shadings) with those driven by observed meteorological forcings (black line). We have also used the historical simulations to select CMIP5 models that are suitable for future projections:

“In fact, 13 CMIP5/ALL/LSM simulations were carried out, and 11 CMIP5 models were selected due to their capability in reproducing the upward trend for flash drought frequency over China for CLM4.5 and VIC, while 7 CMIP5 models were selected for NoahMP (see Supplementary Table 1 for details).” (L354-L357)

Figure 4. Attribution for historical change in flash drought frequency. (a) Observed and simulated anomalies of the number of flash drought events per year averaged over China. Black line shows the ensemble mean of OBS/CLM4.5, OBS/VIC and OBS/NoahMP simulations, red, blue and cyan lines show the land surface model ensemble simulations driven by CMIP5 climate model ensemble simulations with ALL, NAT and GHG forcings, respectively (see Supplementary Table 1 for details). The thick lines are 10-years running means, and the pink, cyan and yellow shadings display the 5%-95% ranges of ALL, NAT and GHG ensemble simulations respectively. (b) The best estimates of the scaling factors (left axis) and attributable increasing trends (right axis) from three-signal (GHG, OANT=ALL-GHG-NAT and NAT) analyses of flash drought changes for the period 1961–2005. Error bars indicate their corresponding 5%–95% uncertainty ranges estimated via Monte Carlo simulations. All the statistics were calculated during the growing seasons (April-September).

d) The last author (Sheffield) published a paper a few years ago arguing no change in droughts globally over the historic record. Yes, these are flash droughts, and he and colleagues investigated more traditional large area persistent droughts. But still, I would expect some interpretation of why the flash drought trends over China are in such strong contrast.

Furthermore, while they do reference the 2012 Sheffield et al paper, it is in the context of an introductory statement about “large uncertainty in future drought projections” – this even though the 2012 paper deals with historic droughts “over the last 60 years” and to my recollection, has nothing to do with future projections.

Response: For the contrast between USA and China, please see our response to your comment #b) above. For the reference, we agree with the reviewer that it is not relevant for future projections. It is related to the uncertainty of drought index where ET parameterization plays an important role. Therefore, we have used the reference to argue the “drought index variety” and revised the manuscript as follows:

“The Fifth Assessment Report of the Intergovernmental Panel on Climate Change concluded that there was a low confidence in detecting and attributing human impact on drought changes since the middle of the 20th century over global land areas due to internal climate variability¹, data scarcity and drought index variety²⁻³, resulting in large uncertainty in future drought projection^{1,4-6}.” (L33-L37)

Taken together, my recommendation is that this paper should not be seriously considered by Nature Communications. Frankly, I don’t think it should have been sent out for review in the first place. The authors would be far better advised to submit their work to a quality disciplinary journal that has a history of publishing in the area, e.g., JHM or J Clim.

Response: We regret that the reviewer did not realize the value of our work, which might be partly due to unclear presentation in our previous version of the manuscript. We hope we have now improved the clarification and addressed the reviewer’s comments above.

In fact, we believe we are the first to propose the new definition of flash drought for detection and attribution of flash drought change, and for future projection of flash drought risk, not just in China, but also in the worldwide literatures. We believe our study has both new insights and broad interests that make it qualified for external review by *Nature Communications*.

Responses to the comments from Reviewer #3

We thank the reviewer for the critical review. The thoughtful comments are valuable and the suggestions make our work more confident. The reviewer's comments are italicized and our responses immediately follow.

1 General Comments

This manuscript presents a new method to characterise flash drought events (i.e., rapid onset, short duration, co-evolving with a heatwave, severe impacts) and to estimate the exposure risk. This method is applied in China because the propensity of semi-humid and humid regions to endure this kind of hazard. The authors claim that this type of event will increase 30% by the second half of this century under a medium socioeconomic challenge scenario increase. Using the fingerprinting approach, the authors attribute that 80% of the expected changes are induced by the increased greenhouse gas concentrations. The subject covered by this study is a highly relevant research topic having broad scientific and public interest, hence, it is highly suitable for this journal. In the present manuscript, however, there are many key points that have to be clarified before publication.

Response: We would like to thank the reviewer for the positive and constructive comments. Please see our detailed responses below.

2 Specific Comments

My major concern with this study is the lack of uncertainty estimates for all major results. It is highly recognised in the literature (see the 2018NCC article authored by the last co-author, ref. nr. 6) that the SMI is highly uncertain due to the large influence of the land surface parameterizations. For this reason is highly recommended to use several LSM to estimate the uncertainty bounds of the SMI estimates (as done in the previous reference). In the present case, the authors only use CLM4.5 (see L.70) to estimate the SMI using CMIP5 forcings. In this way, the authors can estimate the GCM variability but not the LSM variability and hence the results may be over or under-estimated, depending of the bias of the current LSM. Therefore, I strongly suggest the authors to use at least two more LSM for generating a super ensemble. Then, estimate the standard errors of the major results.

Response: We would like to thank the reviewer for this important comment, although it means a

tripled modeling effort. According to the suggestion, we have now augmented the simulations by using another two land surface models (LSMs), VIC and NoahMP. With the two LSMs, we have repeated all simulations twice, and revised all related results throughout the manuscript, including Figs. 2-5 in the main text, and Supplementary Figs. 4-7 and 9-10. All results and their uncertainty estimations are now based on multiple CMIP5 models and multiple LSMs.

As expected by the reviewer, we did overestimate the anthropogenic change without consideration the uncertainty of LSMs in our previous manuscript. The main conclusions in the abstract have now been updated as follows:

“...exposure risk over China will increase by about $21\pm 9\%$ ” (L23 in the revised manuscript)

“...anthropogenic climate change induced by the increased greenhouse gas concentrations accounts for $77\pm 14\%$ of the upward trend of flash drought frequency...” (L27-L28)

Please also see our responses below with more details regarding the inclusion of multiple LSM ensemble simulations.

The link with the heatwaves (part of the definition of flash droughts) is not clear based on what us written. Please elaborate (L147, L240ff). How temperature is taken into account for the definition of a flash drought event? Or is not taken? If not, then the introduction is misleading (L35 ff).

Response: The flash drought events defined in this study do not explicitly consider temperature anomaly, we have now clarified in the revised manuscript as follows:

“The flash droughts defined in this study do not explicitly consider heatwave conditions, but the heat and drought conditions may occur simultaneously for extreme cases, where a bivariate or multivariate (e.g., copula) analysis is needed for investigating their separate contributions and synergistic effect.” (L215-L219)

The reason that we mentioned the heatwaves in the introduction is that flash droughts usually occur with heat extremes, and there is a popular definition of flash drought event by joint distribution of heat and dry anomaly. We have now clarified in the results section as follows:

“Definition and characteristics of flash drought events. A popular definition for flash drought events in the hydroclimate community is based on the joint distribution of positive temperature

anomaly (e.g., heatwave) and soil moisture deficit^{13,20,26}, although the ecohydrological community has different opinions from the perspective of flash drought impact¹⁶. The former essentially defines an event with concurrent heat extreme and dry conditions, but not necessarily a drought event. For a drought event (no matter conventional drought or flash drought), the system should reach a water deficit for a period of time. If a dry anomaly lasts for a very short period (which is common in the previous definitions^{13,20,26}), it may not have any significant impacts on the ecosystem or the society. Therefore, we consider both the rapid decline rate of soil moisture and the dry persistency in this study...” (L75-L84)

I strongly recommend to use a trend-preserving bias correction (see S.Hempel et al. doi:10.5194/esd-4-219-2013.) instead of the equidistant CDF matching method (L312). The method used may have changed the trends of P and T in the CMIP5 models and hence may alter the results of this study.

Response: Thanks for the comment. We have now evaluated the ECDF matching method regarding the trend preservation, and Fig. S3 shows that the method does preserve the trend both for historical simulations and future projections.

At monthly time scale, Hempel et al. (2013) added a constant to the temperature series and multiplied a constant to the precipitation series, which is simple but preserve the trend, although whether we should preserve the trend or not is still under debate (some climate models produce weird or wrong trends). In fact, there are many bias correction method, each method has its advantages and disadvantages. Hempel et al. (2013)’s method is developed for impact studies in a response to different warming levels (e.g., 1.5-degree and 2-degree warming sensitivity studies). Here, we mainly focus on the decadal changes (i.e., results averaged during 2040-2069 or 2070-2099 as compared with those during 1970-1999), so the conclusions in this study are less sensitive to the warming levels. Moreover, the ECDF method basically correct the first and second moment of the monthly variables (mean and standard deviation), which is similar to Hempel et al. (2013) that corrects the mean for temperature, and both mean and standard deviation for precipitation. For the daily time series, we basically keep the variations of CMIP5 models, where a constant is added to the temperature daily series and a constant is multiplied to the precipitation daily series, based on the monthly values before and after bias correction. Therefore, we would like to keep using ECDF matching method in this study.

Figure S3. An example of CMIP5 model data with or without bias correction for historical and future July precipitation (a; mm/day) and temperature (b; mm/day) averaged over China under ALL forcings (1959-2005) and RCP4.5 and RCP8.5 scenarios (2006-2099).

Sampling uncertainty (variability in the model-simulated response) is key for the optimal fingerprinting approach (see Allen and Scott, 2003 10.1007/s00382-003-0313-9). In the present case, I consider that the SMI uncertainty is totally underestimated by the use of a single LSM(CLM4.5). Please investigate, how the inclusion of more LSMs would affect the 80% estimate.

Response: We agree with reviewer, the new estimation result for the GHG contribution based on multiple CMIP5 models and three LSMs is below:

“The best estimates of scaling factors show that only the GHG signal is detectable, with a contribution of $77\% \pm 14\%$ to the increase (Fig. 4b). These results are also consistent with the simulations of flash drought frequency, where GHGs play a dominant role in all anthropogenic forcings (Supplementary Fig. 7).” (L155-L158)

Figure 4. Attribution for historical change in flash drought frequency. (a) Observed and simulated anomalies of the number of flash drought events per year averaged over China. Black line shows the ensemble mean of OBS/CLM4.5, OBS/VIC and OBS/NoahMP simulations, red, blue and cyan lines show the land surface model ensemble simulations driven by CMIP5 climate model ensemble simulations with ALL, NAT and GHG forcings, respectively (see Supplementary Table 1 for details). The thick lines are 10-years running means, and the pink, cyan and yellow shadings display the 5%-95% ranges of ALL, NAT and GHG ensemble simulations respectively. (b) The best estimates of the scaling factors (left axis) and attributable increasing trends (right axis) from three-signal (GHG, OANT=ALL-GHG-NAT and NAT) analyses of flash drought changes for the period 1961–2005. Error bars indicate their corresponding 5%–95% uncertainty ranges estimated via Monte Carlo simulations. All the statistics were calculated during the growing seasons (April–September).

Figure S7. Frequency of flash drought (events/decade) during the growing seasons (April to September) from (a) OBS/LSM ensemble simulations as well as (b)-(d) CMIP5/LSM ensemble simulations with ALL, GHG and NAT forcings, respectively. All statistics are based on the data during 1961-2005.

It would be interesting to know what is the uncertainty contribution of GCMs and LSMs on the flash drought occurrences. This, of course, only can be done by including more LSM in this study.

Response: We have revised the manuscript according to the suggestion as follows:

“The climate data from CMIP5 models were bias corrected (Supplementary Figs. 1-3; see Methods for details) before feeding into the LSMs, and the LSMs included the Community Land Model version 4.5 (CLM4.5)²⁷, the Variable Infiltration Capacity (VIC) model²⁸, and the Noah LSM with multiparameterization options (NoahMP)²⁹. There are considerable uncertainties both

from CMIP5 climate models and LSMs, where the uncertainty could be underestimated if using simulations with a single LSM (Supplementary Fig. 4). As compared with the LSM simulations driven by observed meteorological forcings, the CMIP5/ALL/LSM simulations underestimate the national-averaged flash drought frequency by one event per decade, but it is within the uncertainty of LSMs (Supplementary Fig. 5). Without using LSM ensemble simulations, the uncertainty range for the drought frequency might be underestimated by 24%-47% (Supplementary Fig. 5).” (L114-L124)

Figure S4. Uncertainty ranges of the frequency of flash drought events from CMIP5/ALL/LSM simulations. Here, the uncertainty range was quantified by the 1.645 standard deviation of the frequency (events/decade). There are 29, 11, 11 and 7 realizations for ALL/LSMs (a), ALL/CLM4.5 (b), ALL/VIC (c) and ALL/NoahMP (d) simulations (see Table S1). All statistics are based on the data during 1961-2005.

Figure S5. The same as Figure S4, but for the uncertainty ranges of flash drought frequency averaged over China. Here, OBS/LSMs represent LSM simulations driven by observed meteorological forcings during 1961-2005.

3 Final Remarks

Based on the comments mentioned above and bearing in mind the NCOMMS publishing standards for a research article, I recommend to return this manuscript to the Authors for major revisions.

Response: We would like to thank the reviewer for the positive comments. We hope we have addressed them above.

References:

1. Hempel, S., et al. A trend-preserving bias correction – the ISI-MIP approach. *Earth Syst. Dynam.* **4**, 219–236 (2013).
2. Mo, K. C. & Lettenmaier, D. P. Heat wave flash droughts in decline. *Geophys. Res. Lett.* **42**, 2823–2829 (2015).
3. Mo, K. C., & Lettenmaier, D. P. Precipitation deficit flash droughts over the United States. *J. Hydrometeor.* **17**(4), 1169–1184 (2016).
4. Wang, L., Yuan, X., Xie, Z., Wu, P., & Li, Y. Increasing flash droughts over China during the recent global warming hiatus. *Sci. Rep.* **6**, 30571 (2016).
5. Wang, L. Y. & Yuan, X. Two types of flash drought and their connections with seasonal drought. *Adv. Atmos. Sci.*, **35**(12), 1478–1490 (2018).

Reviewers' comments:

Reviewer #1 (Remarks to the Author):

I presented one major concern in my initial review of this manuscript, that the flash drought definition introduced here did not explicitly consider the duration of the event. To their credit, the authors have clarified their definition, but have not addressed the primary issue. I will try to make my concern more clear because I believe there was some miscommunication.

The current definition requires 1) the pentad-average soil moisture remain below the 40th percentile and decreases to at or below the 20th percentile, and 2) the average decline rate is at or more than 5% per pentad. This captures the "flash", but does not explicitly account for duration. Unless I am missing something, an event in which soil moisture declines from the 35th percentile to the 12th percentile in 2 pentads, and then quickly recovers back to the 40th percentile, would be considered a flash drought, no? In that case, the actual duration of the event would be 10 days, which is too few for drought impacts. This is my issue: the definition should also include a minimal time that the soil moisture remain below the (e.g.,) 30th or 40th percentile before the event is termed a flash drought.

Along this line of argument, the authors continue to refer to flash droughts as "short duration" (line 42) and lasting for "less than one month" (line 55), yet these statements are not supported by previous research. The authors state that the average duration of events identified as flash droughts using their methodology is 15 days, which seems like it should be closer to the minimum duration to be considered a drought. Likely, without an explicit duration requirement included in their definition, many very short (i.e., 5-10 day) events are identified as flash droughts, which will skew the statistics and add considerable uncertainty to the results.

I implore the authors to include an explicit duration requirement in their definition and revise accordingly. Without this, I cannot recommend this paper for publication.

Review towards comments of ref 2: Overall, I think the authors do a solid job responding to the concerns of ref. 2, and revising their manuscript accordingly. With that being said, there are some outstanding issues - mainly with miscommunication or misunderstanding - that should be addressed.

1) Ref. 2 refers to two methodological absences in the paper, one related to the historical base period and the other to GCM bias correction. In both instances, the authors have indeed done the work and/or provided details; however, in both instances the authors should provide more specifics in the results and/or methodology sections. For example, specification of the historical base period (1970-1999) should not be relegated solely to a single figure caption, but instead specified in the experimental design section. It is likely ref. 2's misunderstanding would be replicated by readers if these specifications are not added.

2) I disagree with ref. 2's argument that developing a new flash drought definition is unnecessary. However, I do think the authors should present a more substantial argument for why a new flash drought definition - beyond the flawed Mo and Lettenmaier methodology - is necessary, and how the new definition accounts for these issues. The authors provide some of this in the introduction, but it needs to be thoroughly expanded.

Reviewer #3 (Remarks to the Author):

Comments on the Revised Manuscript

Anthropogenic shift towards higher risk of flash drought over China

By Yuan et al.

In general I am very satisfied with the rebuttal letter and modifications carried out by the authors. It appears that including many LSMs was crucial to refine the conclusions of the manuscript.

Regarding the GCM/LSM contributions, the authors have have provide hints by providing maps of al GCMs with VIC, CLM, etc. This is already interesting, but still not addressing my suggestions fully. I have in mind a map similar to that presented in

Thober, S., et al. (2018), Multi-model ensemble projections of European river floods and high flows at 1.5, 2, and 3 degrees global warming, *Environmental Research Letters*, 13(1), 014003–11, doi:10.1088/1748-9326/aa9e35.

Figure 4. in this paper shows clearly the GCM/HM contributions. The method used in this paper is straightforward and the Authors can easily implement it. I strongly suggest to do it so. This would be a great add-on to this novel subject and important information for new research.

With this minor additions, I consider that this manuscript can be published.

Luis Samaniego

Responses to the comments from Reviewer #1

We thank the reviewer for the critical review. The thoughtful comments have helped improve the manuscript. The reviewer's comments are italicized and our responses immediately follow.

I presented one major concern in my initial review of this manuscript, that the flash drought definition introduced here did not explicitly consider the duration of the event. To their credit, the authors have clarified their definition, but have not addressed the primary issue. I will try to make my concern more clear because I believe there was some miscommunication.

The current definition requires 1) the pentad-average soil moisture remain below the 40th percentile and decreases to at or below the 20th percentile, and 2) the average decline rate is at or more than 5% per pentad. This captures the "flash", but does not explicitly account for duration. Unless I am missing something, an event in which soil moisture declines from the 35th percentile to the 12th percentile in 2 pentads, and then quickly recovers back to the 40th percentile, would be considered a flash drought, no? In that case, the actual duration of the event would be 10 days, which is too few for drought impacts. This is my issue: the definition should also include a minimal time that the soil moisture remain below the (e.g.,) 30th or 40th percentile before the event is termed a flash drought.

Along this line of argument, the authors continue to refer to flash droughts as "short duration" (line 42) and lasting for "less than one month" (line 55), yet these statements are not supported by previous research. The authors state that the average duration of events identified as flash droughts using their methodology is 15 days, which seems like it should be closer to the minimum duration to be considered a drought. Likely, without an explicit duration requirement included in their definition, many very short (i.e., 5-10 day) events are identified as flash droughts, which will skew the statistics and add considerable uncertainty to the results.

I implore the authors to include an explicit duration requirement in their definition and revise accordingly. Without this, I cannot recommend this paper for publication.

Response: We would like to thank the reviewer for the positive and constructive comments.

1) We agree with the reviewer without an explicit duration requirement, many 10-day events are included. We have now revised the definition by specifying the minimum duration of 3-pentad (15 days), and by including those pentads with soil moisture lower than 20th percentile after the onset of flash drought. The revised definition excludes the 10-day events and reduces the flash drought frequency (Fig. 2a), but the mean duration is now between 20-40 days for most areas in China (Fig. 2c). The revised text and figures are as follows:

“Definition and characteristics of flash drought events. To identify a flash drought event, we consider both the rapid decline rate of soil moisture and the dry persistency in this study: 1) the pentad (five-days) mean root-zone (top 1m) soil moisture decreases from above 40th percentile to 20th percentile, with an average decline rate of no less than 5% for each pentad (e.g., June 30-July 14 in Fig. 1); 2) if the declined soil moisture rises up to 20th percentile again, the drought terminates (e.g., July 15-July 19 in Fig. 1); and 3) the drought should last for at least 3 pentads (15 days). The first two criteria describe the onset and recovery stages of a flash drought event. Although the recovery threshold could increase from 20th to 40th percentile, here the 20th percentile is chosen to exclude many events that can last for more than 3-6 months if the 30th or 40th percentile threshold is used, where those events should be regarded as seasonal droughts instead of flash droughts. The third criterion is the minimal time that the soil moisture remains below 40th percentile, which excludes those events that decrease from above 40th percentile rapidly down to 20th percentile within 10 days, but then recover up to 40th percentile suddenly.” (L84-97 in the revised manuscript)

Figure 1. Illustration of the definition of a flash drought event. The black solid line shows five-days mean soil moisture percentile (SM) for a grid point (112.75°E, 25.25°N) during the

2013 southern China flash drought. The dates in the horizontal axis are in the middle of the five-days periods. The blue asterisks and red circles show flash droughts identified by concurrent heat and drought conditions (old; i.e., pentad mean surface air temperature anomaly larger than its standard deviation, and soil moisture percentile lower than 30%) and by the new definition used in this study (new; see text for details), respectively.

Figure 2. Comparison of mean frequency and duration of flash drought events based on two definitions. Mean results for the number of flash drought events per decade (a)-(b), and durations (days) of flash drought events (c)-(d). All statistics are based on the average results from three land surface model (i.e., CLM4.5, VIC, NoahMP) simulations driven by observed meteorological forcings during 1961-2005.

2) The reason for choosing 15 days as the duration threshold is to avoid counting short (10-day) dry spells and to keep those 15-day events that DO have impacts. We have applied the revised definition to the FLUXNET stations to investigate the flash drought impact on vegetation. Figure R1 shows that a 3-pentad flash drought event can have significant adverse effect on gross primary productivity (GPP). In fact, even for a 2-pentad flash drought (which

was identified in our previous definition), we can also find cases with adverse vegetation response (not shown). Therefore, the minimum drought duration may vary for different climate and vegetation regimes, which needs further investigation. Here, we use 15 days as the minimum duration that is considered as a drought.

Figure R1. In-situ soil moisture percentile and gross primary productivity (GPP) anomalies at DE-Tha and US-SRM sites during 1999 and 2011 respectively. DE-Tha is an evergreen needleleaf forest site in Europe (50.96N, 13.56E), and US-SRM is a woody savanna site in USA (31.82N, 110.87W). The shaded cyan areas indicate the identified flash drought events.

3) With the new definition, we have updated all related results and figures, and removed the statements of “short duration”. Please see Figures 1-5, and S4-S8 in the revised manuscript. Although there are changes in the statistics based on the revised flash drought definition by explicitly specifying minimum duration, the main conclusions remain unchanged. The revised texts are as follows:

“Flash droughts refer to a type of droughts that have rapid intensification and severe impacts without sufficient early warning.” (L17-18)

“...and find that the exposure risk over China will increase by about $23\% \pm 11\%$ (more than 40% over southernmost humid provinces) during the middle of this century under a socioeconomic scenario with medium challenge...” (L24-26)

“Optimal fingerprinting shows that anthropogenic climate change induced by the increased greenhouse gas concentrations accounts for $77\% \pm 26\%$ of the upward trend of flash drought frequency...” (L28-30)

“...suggesting a type of drought with rapid onset as accompanied by heat waves...” (L39-40)

“Flash droughts raise the risk of food and water security...” (L43)

“Due to the intrinsic nature of extreme events...” (L50)

“Moreover, projections of future changes are often carried out for droughts at seasonal, annual or longer time scales^{5,6,24}, while flash droughts occur at sub-seasonal time scale from a few weeks to months.” (L53-55)

“Without using LSM ensemble simulations, the uncertainty range for the drought frequency might be underestimated by 21%-44% (Supplementary Fig. 5).” (L131-133)

“On average, China flash drought severity will increase by $18\% \pm 9\%$ and $22\% \pm 12\%$ during 2040-2069 and 2070-2099 respectively...” (L138-139)

“Figs. 3c-3d show that future exposure to flash drought will increase by $23\% \pm 11\%$ and $19\% \pm 16\%$ averaged over China in the middle and end of this century.” (L146-148)

We would like to thank the reviewer again for the critical comment. With the revision on the definition, it will facilitate our follow-up study regarding the flash drought impact.

Responses to the comments from Reviewer #2

We thank the reviewer for the critical review. The reviewer's comments are italicized and our responses immediately follow.

Overall, I think the authors do a solid job responding to the concerns of ref. 2, and revising their manuscript accordingly. With that being said, there are some outstanding issues - mainly with miscommunication or misunderstanding - that should be addressed.

1) Ref. 2 refers to two methodological absences in the paper, one related to the historical base period and the other to GCM bias correction. In both instances, the authors have indeed done the work and/or provided details; however, in both instances the authors should provide more specifics in the results and/or methodology sections. For example, specification of the historical base period (1970-1999) should not be relegated solely to a single figure caption, but instead specified in the experimental design section. It is likely ref. 2's misunderstanding would be replicated by readers if these specifications are not added.

Response: We would like to thank the reviewer for the positive comments. We used different periods for bias corrections and model simulations, drought identification and drought analysis. The baseline period is 1959-2005 for bias corrections and model simulations, 1961-2005 for the calculation of percentiles and identification of flash drought because 1959-1960 data were dropped as model spinup. While for the results showing in Figures 3, 5, S6, S10-S11, we used 1970-1999 as the baseline period according to the previous literature regarding climate change assessment (i.e., comparing droughts in the 21st century with those happened in the end of the 20th century). We have now clarified them in the methodology section as follows:

“To reduce the bias from CMIP5 historical simulations ... during 1959-2005. GHG, NAT and CTL simulations ... during 1959-2005. For the future projection experiments ... changes in historical (1959-2005) and projection (2006-2099) conditions.” (L404-411)

“Although all bias corrections use 1959-2005 as the baseline period for constructing historical distributions and correcting future projections, the soil moisture percentile estimations use 1961-2005 as the baseline period for both historical (1961-2005) attribution and future

(2006-2099) projection of flash droughts, because the soil moisture data during 1959-1960 are dropped as LSM spin-up results.” (L415-419)

2) I disagree with ref. 2's argument that developing a new flash drought definition is unnecessary. However, I do think the authors should present a more substantial argument for why a new flash drought definition - beyond the flawed Mo and Lettenmaier methodology - is necessary, and how the new definition accounts for these issues. The authors provide some of this in the introduction, but it needs to be thoroughly expanded.

Response: Thanks for the positive comments. We clarified the disadvantage of the old definition in the results section in the previous version of the manuscript. According to the suggestion, we have now moved them into the introduction section, and expanded the clarification as follows:

“Quantifying flash drought risk requires an objective identification of a flash drought event. Similar to conventional drought events²⁵, flash drought events are also subject to the processes of onset and recovery. Flash droughts could either evolve into seasonal droughts (e.g., the 2012 central USA summer drought), or terminate independently without connection with droughts at longer time scales. However, existing definitions do not explicitly consider drought intensification processes, or have neglected the recovery stage as well as the severity of a flash drought event^{13,15,20}. A popular definition for flash drought events in the hydro-climate community is based on the joint distribution of positive temperature anomaly (e.g., heatwave) and soil moisture deficit^{13,20,26}, although the eco-hydrological community has different opinions from the perspective of flash drought impact¹⁶. The former essentially defines an event with concurrent heat extreme and dry conditions, but not necessarily a drought event. For a drought event (no matter conventional drought or flash drought), the system should reach a water deficit for a period of time. If a dry anomaly lasts for a very short period (which is common in the previous definitions^{13,20,26}), it may not have any significant impacts on the ecosystem or the society. Moreover, the rapid intensification is also a key feature to distinguish flash droughts from conventional droughts in terms of their physical characteristics and impacts, and the identification of the severity and different drought stages facilitates early warning and risk assessment during the evolution of flash droughts. Here, we propose a new flash drought

definition based on soil moisture that can capture both “flash” (rapid intensification of a drought condition, e.g., rapid decline in soil moisture) and “drought” (under a certain soil moisture threshold for a period of time) conditions.” (L57-78)

Responses to the comments from Reviewer #3

We thank the reviewer for the critical review. The thoughtful comments are valuable and the suggestions make our work more confident. The reviewer's comments are italicized and our responses immediately follow.

In general I am very satisfied with the rebuttal letter and modifications carried out by the authors. It appears that including many LSMs was crucial to refine the conclusions of the manuscript.

Regarding the GCM/LSM contributions, the authors have have provide hints by providing maps of al GCMs with VIC, CLM, etc. This is already interesting, but still not addressing my suggestions fully. I have in mind a map similar to that presented in

Thober, S., et al. (2018), Multi-model ensemble projections of European river floods and high flows at 1.5, 2, and 3 degrees global warming, Environmental Research Letters, 13(1), 014003–11, doi:10.1088/1748-9326/aa9e35.

Figure 4. in this paper shows clearly the GCM/HM contributions. The method used in this paper is straightforward and the Authors can easily implement it. I strongly suggest to do it so. This would be a great add-on to this novel subject and important information for new research.

With this minor additions, I consider that this manuscript can be published.

Response: We would like to thank the reviewer for the positive comments. According to the suggested uncertainty estimation method³⁰, we have now included a figure to show the contribution of uncertainty for GCM/LSM for the projection of flash drought frequency and duration, and revised the manuscript as follows:

“In the flash drought projection, the uncertainty³⁰ from CMIP5 models is larger than that from LSMs, especially over northern China (Supplementary Fig. 7).”

Figure S7. The ratio between the Global Climate Model (GCM; 11 CMIP5 models used in this study) contribution and LSM contribution to the uncertainty ranges for the changes in flash drought frequency and duration shown in Figure S6. The values larger than 1 suggest that the uncertainty from GCMs is larger than that from LSMs.

References:

- 30. Thober, S., et al. Multi-model ensemble projections of European river floods and high flows at 1.5, 2, and 3 degrees global warming. *Environ. Res. Lett.* **13**, 014003 (2018).

REVIEWERS' COMMENTS:

Reviewer #1 (Remarks to the Author):

The authors have adequately addressed my previous concerns, and I recommend the manuscript be accepted for publication.

Reviewer #3 (Remarks to the Author):

Comments on
"Anthropogenic shift towards higher risk of flash drought over China"
by X. Yuan et al.

This is the second review of this manuscript. I would like to congratulate the authors for the revised version. It has definitely been improved and reads well. The topic is at the forefront of the state-of-the-art on this subject and hence suitable for publishing in Nature Comm.

I am satisfied with the quality of the response to my suggestions. The inclusion of the definition of a flash drought suggested by another reviewer, I find also very welcome. Consequently, I consider that this manuscript could be published as it is.

LS